# Methods for Drainage of Distal Malignant Biliary Obstruction after ERCP Failure: A Systematic Review and Network Meta-Analysis

**DOI:** 10.3390/cancers14133291

**Published:** 2022-07-05

**Authors:** Antonio Facciorusso, Benedetto Mangiavillano, Danilo Paduano, Cecilia Binda, Stefano Francesco Crinò, Paraskevas Gkolfakis, Daryl Ramai, Alessandro Fugazza, Ilaria Tarantino, Andrea Lisotti, Pietro Fusaroli, Carlo Fabbri, Andrea Anderloni

**Affiliations:** 1Gastroenterology Unit, Department of Surgical and Medical Sciences, University of Foggia, 71122 Foggia, Italy; antonio.facciorusso@unifg.it; 2Gastroenterology and Digestive Endoscopy Unit, Department of Medicine, The Pancreas Institute, University Hospital of Verona, 37100 Verona, Italy; stefanofrancesco.crino@aovr.veneto.it; 3Gastrointestinal Endoscopy Unit, Humanitas Mater Domini, Via Gerenzano 2, 21053 Castellanza, Italy; benedetto.mangiavillano@mc.humanitas.it (B.M.); danilo.paduano@materdomini.it (D.P.); 4Gastroenterology and Digestive Endoscopy Unit, Forlì-Cesena Hospitals, AUSL Romagna, 47121 Forli, Italy; carlo.fabbri@auslromagna.it; 5Department of Gastroenterology, Hepatopancreatology, and Digestive Oncology, CUB Erasme Hospital, Université Libre de Bruxelles (ULB), 1050 Brussels, Belgium; paraskevas.gkolfakis@erasme.ulb.ac.be; 6Gastroenterology and Hepatology, University of Utah Health, Salt Lake City, UT 801385, USA; daryl.ramai@hsc.utah.edu; 7Digestive Endoscopy Unit, Humanitas Clinical and Research Center-IRCCS, Via Manzoni 56, Rozzano, 20089 Milano, Italy; alessandro.fugazza@humanitas.it (A.F.); a.anderloni@smatteo.pv.it (A.A.); 8Endoscopy Service, Department of Diagnostic and Therapeutic Services, IRCCS-ISMETT, 90100 Palermo, Italy; itarantino@ismett.edu; 9Gastroenterology Unit, Hospital of Imola, University of Bologna, 0039051 Bologna, Italy; lisotti.andrea@gmail.com (A.L.); pietro.fusaroli@unibo.it (P.F.)

**Keywords:** EUS, surgery, cancer, metastasis, stent

## Abstract

**Simple Summary:**

With the widespread use of EUS-guided procedures, several methods are available in order to achieve biliary drainage when endoscopic retrograde cholangiopancreatography (ERCP) fails. Together with the well established role of percutaneous trans-hepatic biliary drainage (PTBD) and surgical hepaticojejunostomy, EUS-choledochoduodenostomy (EUS-CD) and EUS-hepaticogastrostomy (EUS-HG) have provided good results to date, representing valuable alternatives. However, no definite indications have been provided about which should be the best way of drainage. In this network meta-analysis, we compared all these techniques, showing how, considering the available studies, none of these methods seems to be superior to another, although PTBD seems to have a slightly higher rate of adverse events. So, when dealing with patients affected by distal malignant biliary obstruction (DMBO) and when ERCP fails, all these methods seem to be equally effective, although possibly EUS-guided approaches could be less invasive and affected by fewer adverse events.

**Abstract:**

There is scarce evidence on the comparison between different methods for the drainage of distal malignant biliary obstruction (DMBO) after endoscopic retrograde cholangiopancreatography (ERCP) failure. Therefore, we performed a network meta-analysis to compare the outcomes of these techniques. We searched main databases through September 2021 and identified five randomized controlled trials. The primary outcome was clinical success. The secondary outcomes were technical success, overall and serious adverse event rate. Percutaneous trans-hepatic biliary drainage was found to be inferior to other interventions (PTBD: RR 1.01, 0.88–1.17 with EUS-choledochoduodenostomy (EUS-CD); RR 1.03, 0.86–1.22 with EUS-hepaticogastrostomy (EUS-HG); RR 1.42, 0.90–2.24 with surgical hepaticojejunostomy). The comparison between EUS-HG and EUS-CD was not significant (RR 1.01, 0.87–1.17). Surgery was not superior to other interventions (RR 1.40, 0.91–2.13 with EUS-CD and RR 1.38, 0.88–2.16 with EUS-HG). No difference in any of the comparisons concerning adverse event rate was detected, although PTBD showed a slightly poorer performance on ranking analysis (SUCRA score 0.13). In conclusion, all interventions seem to be effective for the drainage of DMBO, although PTBD showed a trend towards higher rates of adverse events.

## 1. Introduction

Patients with malignant biliary obstruction are typically treated with endoscopic retrograde cholangiopancreatography with stent placement. However, failure of the procedure occurs in about 5–7% of patients in patients with altered anatomy (i.e., Whipple intervention, Roux-en-Y gastric bypass, Billroth II surgery), as well as gastric outlet obstruction, periampullary diverticula, or luminal malignant obstruction [1]. In these cases, surgical bypass with percutaneous transhepatic biliary drainage (PTBD) represents an alternative to failed ERCP.

However, these treatments are associated with increased morbidity and adverse events ranging up to 33% [2]. In fact, common adverse events after PTBD placement are catheter dislodgement, recurrent infection, acute cholangitis, bleeding, bile leaks, pneumothorax, and subjective discomfort due to external drainage [3].

In recent years, EUS-guided biliary drainage (EUS-BD) has been increasingly offered when standard ERCP fails. A recent meta-analysis reported cumulative technical success and post-procedure adverse events of 90% and 17%, respectively [4].

With the development of lumen apposing metal stents (LAMS) designed for endoscopic ultrasounds, EUS-BD has evolved as a therapeutic modality. Recently, the development of electrocautery mounted on the tip of LAMS (EC-LAMS) has expanded the armamentarium for endoscopists. This allows the device to be deployed without having to use a 19G needle, guidewire, or the need for prior dilation with a cystotome.

With the rise of EUS-BD, several approaches have been developed, including EUS-guided choledochoduodenostomy (EUS-CD), EUS-guided hepaticogastrostomy (EUS-HG), EUS-guided rendezvous, and EUS-guided antegrade transpapillary drainage [5,6]. Of these techniques, EUS-CD and EUS-HG are the two main approaches used in biliary drainage. Several studies and met a-analysis has been published over the years comparing both techniques, showing comparable outcomes with respect to clinical and technical success rates, as well as adverse outcomes and mortality [5,7,8,9,10].

In a meta-analysis that included nine studies (of which three were randomized trials), EUS-BD was compared to PTBD [11]. The study reported that EUS-BD had higher clinical success than PTBD, as well as fewer post-procedure adverse events, and a lower rate of reintervention; EUS-BD was found to be more cost-effective. Although EUS-BD seems to provide advantages when compared to PTBD, the European Guidelines do not provide strong recommendations on how to approach a biliary drainage in case of ERCP failure, stating that EUS-BD should be restricted to those cases in a palliative setting [12]. To date, indeed, the strategy of choice when ERCP fails is still extremely heterogenous among countries and centers, depending on the available facilities, expertise and devices. However, with the increasing use of chemotherapies in this setting of patients, it will be mandatory in the near future to tailor all the strategies needed to maintain an adequate biliary drainage and improve the quality of life, resulting from a deep comprehension of the available strategies to achieve it. However, an overall comparative assessment among the different interventions for DMBO after ERCP failure is still lacking.

Unlike a pairwise meta-analysis, a network meta-analysis is capable of comparing the efficiency of several interventions and pool data from randomize clinical trials. To this end, network meta-analysis provides comparative outcome data, which informs practice guidelines.

We aim to perform a systematic review and network meta-analysis, comparing the efficacy of EUS techniques used for the drainage of distal malignant biliary obstruction (DMBO) following failed ERCP. In this study, our primary outcome was clinical success as well as adverse events. Quality of evidence was appraised using the Grading of Recommendations Assessment, Development and Evaluation (GRADE) criteria for network meta-analysis.

## 2. Materials and Methods

### 2.1. Selection Criteria

Studies included in this meta-analysis (CRD42022337046) were parallel group RCTs, which met the following inclusion criteria: (a) patients included subjects with DMBO and failure of ERCP to attain drainage; (b) interventions and comparators included EUS-CD with use of self-expandable metal stents (SEMS); EUS-HG with use of SEMS; PTBD; surgical hepaticojejunostomy; (c) outcomes included studies that had to report technical success, clinical success, and postprocedure adverse events. We excluded observational or non-randomized studies, and trials published only as conference abstracts.

### 2.2. Search Strategy, Data Abstraction and Risk of Bias Assessment

Using several electronic databases, a comprehensive search was undertaken from inception to September 2021 using no language restrictions. Databases searched included Ovid Epub, MEDLINE, In-Process and other non-indexed citations, Ovid MEDLINE, Ovid EMBASE, Ovid Cochrane Central Register of Controlled Trials, Ovid Cochrane Database of Systematic Reviews, Scopus and Web of Science. One author with input from other investigators designed and conducted the search strategy. We used the following general search strategy, then repeated it according to the specific syntax for the query of each database: (biliarydrainage[MeSH Terms]) AND (stent[MeSH Terms]) OR (obstruction[MeSH Terms]) AND (trial).

Using a standardized form, study data were extracted by two investigators (independently); any discrepancies were resolved by a third investigator. The quality of the included randomized clinical trials was rated by two investigators using the Cochrane Risk of Bias Tool [13]. A third reviewer evaluated and addressed any disagreements.

### 2.3. Outcomes Assessed

The primary outcome of interest was clinical success, defined mainly as the resolution of biliary obstruction clinically as well as by laboratory parameters or decrease in bilirubin by 50% at 7 days. Other outcomes were technical success, defined as successful stent placement as determined endoscopically, radiographically, or surgically, and adverse event rate.

The severity of adverse events was graded as mild, moderate, severe, or fatal according to the American Society for Gastrointestinal Endoscopy (ASGE) classification [14].

### 2.4. Statistical Analysis

A multivariate random effects meta regression using a consistency model was used to conduct the network analysis as explained by Ian White [15]. From the network, we used a frequentist approach to provide a point estimate expressed as risk ratio (RR) along with 95% confidence intervals (CI) from the distribution frequency of the estimate. Using a node splitting technique, the network consistency was assessed by comparing the direct and indirect estimates for each comparison.

We assessed statistical heterogeneity using the I^2^ statistic, with values over 50% indicating significant heterogeneity, and small study effects were assessed by examining funnel plot asymmetry.

All network meta-analyses were performed using R software (Cochrane Collaboration, Copenhagen, Denmark; *netmeta* package).

An intention to treat analysis was used for all analyses. Interventions for achieving the primary and the secondary outcomes were ranked by their surface under the cumulative ranking (SUCRA) value. SUCRA values range between 0 when a treatment is certainly the worst, and 1 when a treatment is certainly the best [16]. Thus, higher scores correspond to higher rankings for achieving clinical/technical success or preventing adverse event occurrence.

Separate network models were built for overall adverse event and moderate-severe adverse event rates.

### 2.5. Quality of Evidence

GRADE criteria were used to rate the quality of evidence derived from the pairwise and network meta-analysis [17]. Using this approach, RCTs are deemed to have the highest quality of evidence and can be down rated based on bias, imprecision, or heterogeneity in the data. To this end, studies can be down rated to moderate, low, and very low quality. Starting at the lowest rating of the two pairwise estimates (that contribute as first-order loops to the indirect estimate), the rating of indirect estimates can be further down rated for imprecision or intransitivity (dissimilarity between studies in terms of clinical or methodological characteristics).

## 3. Results

### 3.1. Characteristics of Included Studies

From the 143 unique studies identified using the search strategy, 5 RCTs [10,18,19,20,21] were identified and included in the network meta-analysis (Figure 1). Study characteristics and main demographic data are reported in Table 1. Overall, these 5 trials included 217 patients in total, who were well-balanced in terms of baseline characteristics.

The trial by Lee et al. [19] compared PTBD versus EUS-BD reporting separately data concerning EUS-CD and EUS-HG, whereas another RCT compared EUS-CD versus PTBD [18]; two RCTs compared EUS-CD versus EUS-HG [10,21], and one RCT compared surgical hepaticojejunostomy versus EUS-CD [20].

Pancreatic adenocarcinoma was the most common etiology for biliary obstruction in the included trials. The majority of treated patients were male with a mean age ranging from 63.4 to 72.5 years.

Clinical success was defined by the decrease in bilirubin levels to <50% as compared to baseline within 7 days in 3 RCTs [14,15,17], within 14 days in 1 RCT [10], with no specific timespan in 1 RCT [18].

Risk of bias assessment was performed in the context of the primary outcome and, overall, the studies were felt to be at a moderate risk of bias, mainly due to performance bias. Overall and study level quality assessments are summarized in Appendix A, respectively.

### 3.2. Clinical Success Rate

In the network meta-analysis, no inconsistency was observed between the results of direct and indirect comparison. Figure 2 shows the networks of trials, considering the treatments defined as described above. The main results of the network meta-analysis are reported in Table 2.

None of the treatments were superior to PTBD (RR 1.01, 0.88–1.17 with EUS-CD; RR 1.03, 0.86–1.22 with EUS-HG; RR 1.42, 0.90–2.24 with surgery, Table 2 and Figure 3a).

Similarly, the comparison between EUS-HG and EUS-CD was not significant (RR 1.01, 0.87–1.17) and surgery was not significantly superior to the other treatments (RR 1.40, 0.91–2.13 with EUS-CD and RR 1.38, 0.88–2.16 with EUS-HG).

In then ranking analysis, surgery showed a slightly increased SUCRA score (0.54) as compared to the other treatments, but with no evidence of significant superiority (Table 3).

We did not find any evidence of small study effects based on funnel plot asymmetry (data not shown) and there was no significant difference between the direct and indirect estimates in the closed loops that allowed the assessment of network coherence.

### 3.3. Secondary Outcomes

As reported in Table 2, no significant difference in any of the comparisons between endoscopic or surgical methods versus PTBD (Figure 3b) or between different methods was detected (RRs ranging from 0.35, 0.05–2.31 in the comparison between surgery versus PTBD to 0.85, 0.39–1.82 in the comparison between EUS-HG and EUS-CD). As a consequence, PTBD showed a slightly poorer performance in terms of safety profile in the ranking analysis (SUCRA score 0.13), whereas the other treatments presented higher scores (Table 3).

These findings were confirmed in the sensitivity analysis restricted to moderate and severe adverse events, where again PTBD gained the lowest SUCRA score in treatment ranking (Appendix A).

As reported in the Appendix A, the most frequent adverse event was bleeding, which was significantly more frequent with PTBD (pooled incidence 9.1%, 0.6–17.6% vs. 3.4%, 0–7.3% with EUS-CD and 2.9%, 0–6.9% with EUS-HG; *p* = 0.02). No significant difference was registered with regard to perforation, stent migration or incidence of acute pancreatitis.

The technical success rate was similar across all the tested interventions and SUCRA scores were not significantly different, ranging from 0.31 with PTBD to 0.54 with surgery (Appendix A).

### 3.4. Quality of Evidence

The overall body of evidence was rated down due to the risk of performance bias and imprecision, whereas there was no inconsistency, indirectness, or publication bias for any of the comparisons. Therefore, based on the network meta-analysis, the low quality of evidence supported the comparisons of the different interventions (Table 2).

## 4. Discussion

Until a few years ago, surgical drainage or PTBD were the only possible procedures for biliary drainage following ERCP failure. However, PTBD is burdened by a relatively high rate of morbidity and surgery is not commonly feasible due to poor performance status. EUS-BD, first published by Giovannini et al. in 2001 [22], is performed worldwide with favorable technical success and post-procedure adverse event rates. Previous meta-analyses showed that EUS-BD was associated with better clinical success and fewer post-procedure complications than PTBD [8]. Another report indicated comparable performance rates between EUS-CD and EUS-HG [7]. However, an overall comparative assessment among the different interventions for DMBO after ERCP failure is still lacking. Moreover, previous meta-analyses [5,7,8,9,10,11] included both RCTs and retrospective studies, hence the pressing need of a network meta-analysis based on the combination of direct and indirect evidence to gather comparative data to inform clinical guidelines. Therefore, through a network meta-analysis of five RCTs, we made several key observations. First, none of the included interventions were superior to PTBD in terms of success rate and safety profile. This result was in contrast with the above cited meta-analysis [11]. However, the authors of that meta-analysis acknowledged that clinical success was similar for both procedures in the subgroup analysis restricted only to RCTs, whereas clinical success was higher in EUS-BD based on observational studies [11].

It should be noted that all the included RCTs testing EUS-BD were published before the implementation in the clinical practice of LAMS; hence, our findings are not applicable to these newer devices, which indeed showed very promising results in this setting [23,24,25]. For a recent large multicenter study, on 239 patients that underwent EUS-CD using LAMS, technical success was reported in 93.3% of the cases, while clinical success was obtained in 96.2% of the patients in which technical success was achieved [26]. Although there was no significant difference in the rate of adverse events between EUS-BD and PTBD, the latter showed a trend towards higher complication rates in the ranking analysis (SUCRA score 0.13). In particular, pooled rates of bleeding were significantly higher with PTBD (pooled incidence 9.1%, vs. 3.4% with EUS-CD and 2.9% with EUS-HG; *p* = 0.02). Again, these results are in keeping with the current literature. It should be noted that the preliminary results from a recent RCT comparing the two types of procedures reported a strikingly increased rate of adverse events in the PTBD group, because the study stopped enrolling patients in the percutaneous arm [27]. The relatively small sample size and low number of RCTs possibly prevented a significant difference being found in our meta-analysis and further studies are needed to confirm whether this trend would reach the significance threshold.

Secondly, the comparison between EUS-HG and EUS-CD was not significant in any of the outcomes assessed, thus confirming the results of a previous meta-analysis [7]. However, as commented above, our results will likely need to be updated, considering the increasing evidence on the use of LAMS for EUS-CD [26], as well the widespread use of new dedicated stents for EUS-HGS [28]. Furthermore, other EUS-guided interventions could not be tested in our network meta-analysis due to the lack of RCTs, namely EUS-guided rendezvous transpapillary drainage, where the guidewire is placed in the biliary duct under EUS guidance and ERCP biliary cannulation is then reattempted by using the EUS-placed guidewire, and the antegrade stent placement characterized by the guidewire placement under EUS-guidance through the dilated left intra-hepatic duct [29]. Therefore, our meta-analysis could assess the performance only of the two most common techniques for EUS-BD, confirming the comparability of the outcomes between EUS-CD and EUS-HG.

Finally, surgical hepaticojejunostomy showed a favorable efficacy and safety profile. However, the limited evidence based on a single RCT [20], the invasiveness of the procedure, and the fact that these patients are commonly unsuitable for surgery due to comorbidities or poor performance status, prevent the use of surgery on a large scale.

All of the above commented findings were informed by the low quality of evidence due to the risk of performance bias in the literature and imprecision of the results. Of note, the risk of performance bias is very common in the endoscopic literature due to the unblinded design of the endoscopic RCTs, where blinding of the operator is usually not feasible. Nevertheless, using GRADE criteria, we believe that our critical appraisal of the quality of evidence can serve to inform clinicians and future guidelines. There are certain limitations that merit further discussion. First, the limited number of head-to-head trials yielded low quality of evidence for different endoscopic methods. Second, due to conceptual heterogeneity, network meta-analyses may be subject to misinterpretation. This may be the result of considerable differences in participants, interventions, background treatments, and outcome evaluation. However, we restricted our literature search only to patients with DMBO and that had experienced a previous ERCP failure, hence a relatively homogeneous subset of patients. Moreover, the definition of the outcome was similar across the included RCTs and where both direct and indirect evidence was available, we observed no difference in the effect estimates, supporting the transitivity assumption in our network meta-analysis. Third, as already commented above, some specific EUS-guided procedures, in particular new electrocautery-enhanced LAMS, could not be assessed due to the lack of RCTs. This represents a major limitation of the current literature in this field and further research is absolutely needed in this regard.

Concerning the limitations of individual studies, we acknowledge that most of them had a small sample size and others were hampered by different technical details, such as the size and kind of the stent used, although all the stents were classified as SEMS in the included RCTs. Finally, cost considerations were beyond the scope of this work and were not analyzed.

Despite all these limitations, our study highlights how biliary drainage can be safely and effectively performed in different ways, considering the available expertise and facilities. The complementarity between the techniques nowadays is progressively increasing, reflecting the need a new idea of “biliopancreatic procedure” that can overcome a mere “technical-based” concept of treatment to form a “goal-based” one. In parallel, a new way of informed consent should be proposed and adopted, especially in this setting of patients, considering all the options of treatment that can be offered, possibly in a single session [30].

## 5. Conclusions

This is the first network meta-analysis to assess the methods applied in the drainage of DMBO when ERCP fails. While robust GRADE methodology demonstrated the low quality of evidence, we observed that none of the tested methods were superior to the others, although PTBD showed a trend towards increased rates of adverse events; in particular, it was associated with a significant increase in bleeding events.

Further RCTs are needed to confirm our results and to test the alternative EUS-guided methods for biliary drainage.

## Figures and Tables

**Figure 1 cancers-14-03291-f001:**
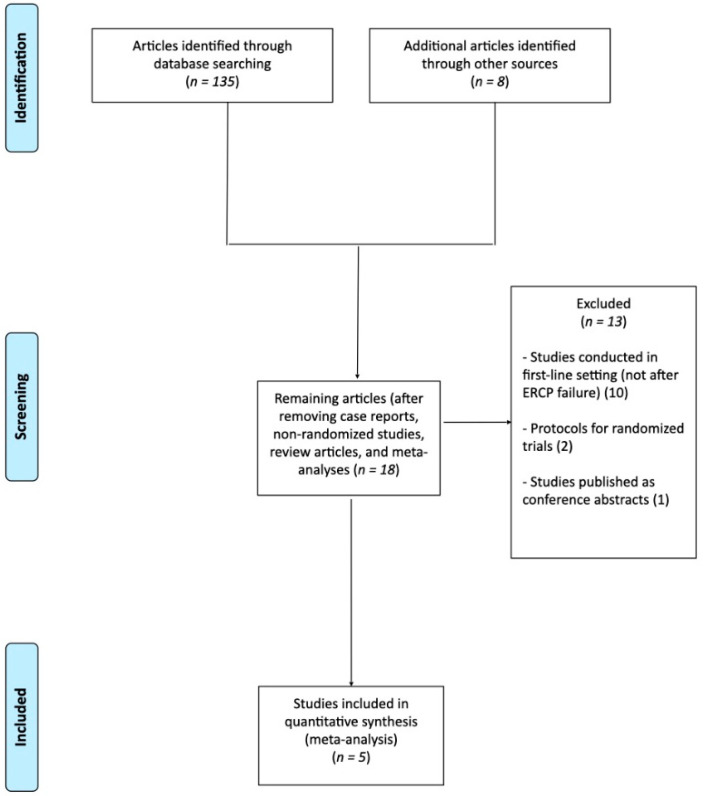
Flow chart of the included studies.

**Figure 2 cancers-14-03291-f002:**
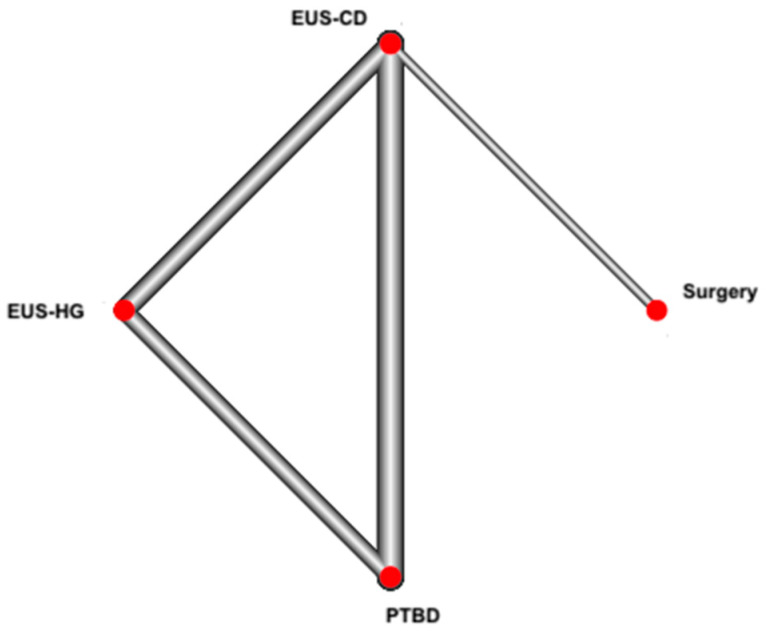
Network of the interventions tested in the included trials.

**Figure 3 cancers-14-03291-f003:**
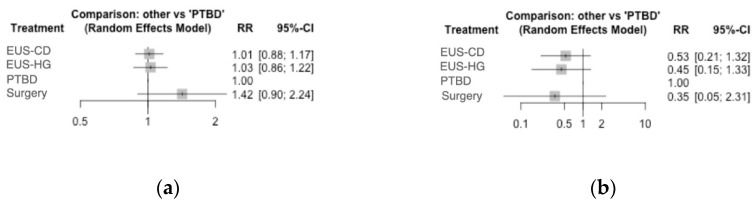
Forest plots of the network meta-analysis comparing different interventions to percutaneous trans-hepatic biliary drainage in terms of (**a**) clinical success rate and (**b**) adverse event rate.

**Table 1 cancers-14-03291-t001:** Baseline characteristics of the included trials.

Study, Year	Location; Time Period	Neoplasia	Intervention (N)	Control (N)	Max Diameter Common Bile Duct (mm)	Age	Gender, Male	Definition of Clinical Success
**EUS-choledochoduodenostomy vs. PTBD**								
Artifon, 2012 [18]	Brazil; 2007–2011	Pancreatic adenocarcinoma 16, Ampullary adenocarcinoma 1, Hematologic neoplasia 2, Cholangiocarcinoma 2, Metastasis 3, Gastric carcinoma 1.	EUS-CD with SEMS (13)	PTBD with SEMS (12)	EUS-CD 13.7 (9 to 28) PTBD 11.9 (8 to 23)	EUS-CD 63.4 ± 11.1 PTBD 71 ± 11.9	EUS-CD 9 (69.2%) PTBD 8 (66.6%)	Improvement of clinical symptoms and decrease in liver enzymes
Lee, 2016 (I) [19] *	Korea; 2014–2015	Cholangiocarcinoma 21, Pancreatic adenocarcinoma 24, Gallbladder carcinoma 8, Ampullary adenocarcinoma 1, Metastasis 4, Gastric carcinoma cancer 5, Duodenal carcinoma 3	EUS-CD with SEMS (8)	PTBD with SEMS (32)	EUS-CD 11.2 ± 4.38 PTBD 12.6 ± 6.18	EUS-CD 66.5 (40–83) PTBD 68.4 (52–82)	EUS-CD NR PTBD 24 (75%)	Decrease in bilirubin level to less than 50% of baseline within 7 days, or less than 75% within 30 days.
**EUS-choledochoduodenostomy vs. Surgical hepaticojejunostomy**								
Artifon, 2015 [20]	Brazil; 2011–2013	NR	EUS-CD with SEMS (16)	Surgical hepaticojejunostomy (16)	EUS-CD 20 Hepaticojejunostomy 20	EUS-CD 65 ± 12.2 Hepaticojejunostomy 68.1 ± 19.5	EUS-CD 7 (43.7%) Hepaticojejunostomy 7 (43.7%)	Decrease in bilirubin level to less than 50% of baseline within 7 days
**EUS-choledochoduodenostomy vs. EUS-hepaticogastrostomy**								
Minaga, 2019 [10]	Japan; 2013–2016	Pancreatobiliary cancer 41, Other 6	EUS-CD with SEMS (23)	EUS-HG with SEMS (24)	NR	EUS-CD 73 (41–83) EUS-HG 72.5 (46–88)	EUS-CD 10 (43.4%) EUS-HG 14 (58.3%)	Decrease in bilirubin level to less than 50% of baseline within 14 days
Artifon, 2015 [21]	Brazil; 2010–2013	Pancreatic adenocarcinoma 33, Metastatic adenopathy 8, Ampullary carcinoma 4, Neuroendocrine tumor 2, Gallbladder cancer 1, Duodenal carcinoma 1	EUS-CD with SEMS (24)	EUS-HG with SEMS (25)	EUS-CD 22.23 ± 4.09 EUS-HG 21.43 ± 4.88	EUS-CD 65.7 (15–74) EUS-HG 66.25 (14–28)	EUS-CD 11 (45.8%) EUS-HG 11 (44%)	Decrease in bilirubin level to less than 50% of baseline within 7 days
**EUS-hepaticogastrostomy vs. PTBD**								
Lee, 2016 (II) [19] *	Korea; 2014–2015	Cholangiocarcinoma 21, Pancreatic adenocarcinoma 24, Gallbladder carcinoma 8, Ampullary adenocarcinoma 1, Metastasis 4, Gastric carcinoma cancer 5, Duodenal carcinoma 3	EUS-HG with SEMS (24)	PTBD with SEMS (32)	NR PTBD 12.6 ± 6.18	EUS-HG 66.5 (40–83) PTBD 68.4 (52–82)	EUS-HG NR PTBD 24 (75%)	Decrease in bilirubin level to less than 50% of baseline within 7 days, or less than 75% within 30 days.

Abbreviations: EUS-CD, EUS-choledochoduodenostomy; EUS-HG, EUS-hepaticogastrostomy; NR, not reported; PTBD, percutaneous trans-hepatic biliary drainage. * This study included two subgroups of EUS-guided biliary drainage, namely EUS-CD and EUS-HG. Demographical data refer to the overall cohort.

**Table 2 cancers-14-03291-t002:** Results of the network meta-analysis concerning clinical success rate and adverse event rate.

	Clinical Success Rate		Adverse Event Rate	
	Risk Ratio (95% CI)	Quality of Evidence	Risk Ratio (95% CI)	Quality of Evidence
**All treatments vs. PTBD**				
EUS-CD	1.01 (0.88–1.17)	Low	0.53 (0.21–1.32)	Low
EUS-HG	1.03 (0.86–1.22)	Low	0.45 (0.15–1.33)	Low
Surgery	1.42 (0.90–2.24)	Low	0.35 (0.05–2.31)	Low
**vs. EUS-CD**				
EUS-HG	1.01 (0.87–1.17)	Low	0.85 (0.39–1.82)	Low
Surgery	1.40 (0.91–2.13)	Low	0.66 (0.12–3.46)	Low
**vs. EUS-HG**				
Surgery	1.38 (0.88–2.16)	Low	0.78 (0.12–4.83)	Low

Abbreviations: EUS-CD, endoscopic ultrasound choledochoduodenostomy; EUS-HG, endoscopic ultrasound hepatico-gastrostomy; PTBD, percutaneous trans-hepatic biliary drainage.

**Table 3 cancers-14-03291-t003:** SUCRA score ranking of the tested interventions.

Clinical Success Rate		Adverse Event Rate	
Surgery	0.54	Surgery	0.63
EUS-HG	0.42	EUS-HG	0.57
EUS-CD	0.34	EUS-CD	0.48
PTBD	0.31	PTBD	0.13

Abbreviations: EUS-CD, endoscopic ultrasound choledochoduodenostomy; EUS-HG, endoscopic ultrasound hepatico-gastrostomy; PTBD, percutaneous trans-hepatic biliary drainage.

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
