# Peer review of "Methods for Drainage of Distal Malignant Biliary Obstruction after ERCP Failure: A Systematic Review and Network Meta-Analysis"

_cancers, 2022, doi:10.3390/cancers14133291_

Round 1

Reviewer 1 Report

I read with great interest your gmanuscript on the most appropriate technique for biliary drainage in distal malignant biliary obstruction after a failed ERCP. This is a problem that centers performing advanced ERCP face from time to time. In practice there are two options: PTBD (percutaneous biliary drainage) and recently EUS - biliary drainage (mainly, choledochoduodenostomy and hepaticogastrostomy).

The problem with your meta-analysis iresults from the low number of studies (5), with small number of patients. Moreover the heterogeneity of the studies are another major problem.

Although the conclusions seems totally acceptable and common sense, for an interventional endoscopist (as myself which perform ERCP and interventional EUS and face this problem from time to time), do you thonk that a meta-analysis could solve the problems that yopu report in the limitations. Moreover, what would be interesting, would be to include the new LAMS (with cautery in the tip) in this evaluations.

Author Response

The reviewer is exactly right in pointing out the low number of studies with limited sample size and these aspects were commented among the limitations to our study in the Discussion. However, as reported in the Discussion, we restricted our literature search only to patients with distal malignant biliary obstruction and that experienced a previous ERCP failure, hence a relatively homogeneous subset of patients. So we preferred to run our analysis in a smaller but homogeneous group of patients in order to provide the reader with reliable and clinically sounded conclusions.

Although all network meta-analysis are subjected to a sort of conceptual heterogeneity due to the presence of indirect comparisons (as commented among the limitations to the study), it should be noted that the studies were homogeneous (as reported above) and the analysis was characterized by the lack of heterogeneity. Moreover, as reported in the Results section, “there was no significant difference between direct and indirect estimates in closed loops that allowed assessment of network coherence”. As a consequence, as clearly stated in the chapter “quality of evidence” of the Results section, there was no evidence of inconsistency (heterogeneity) in the analysis.

Most of the limitations listed in the Discussion were partially addressed, as commented at page 10: “However, we restricted our literature search only to patients with DMBO and that experienced a previous ERCP failure, hence a relatively homogeneous subset of patients. Moreover, the definition of the outcome was similar across the included RCTs and where both direct and indirect evidence was available, we observed no difference in effect estimates, supporting the transitivity assumption in our network meta-analysis.” On the other hand, based on the reviewer’s comment, we highlighted in the Discussion as a major limitation the lack of RCTs testing new LAMS with cautery in the tip.

Reviewer 2 Report

The manuscript represents an interesting systematic review related to methods for drainage of distal malignant biliary obstruction after ERCP failure. 

The introductory part could be developed and more references added. 

The discussion part should also be developed. 

Overall, more references related to the subject are needed,

Author Response

I would like to thank the reviewer 2 for the considerations and contribution that led to ameliorate the manuscript. 

In the introductory part could be developed and more references added. The introductory part has been developed and references has been added.

The discussion part has also be developed and reference has been added.

I hope that the manuscript in the current form could be suitable for publication.